# Disentangling the Roles of Curation, Data-Augmentation and the Prior in the Cold Posterior Effect

**Lorenzo Noci**[*]
Dept of Computer Science
ETH Zürich
lorenzo.noci@inf.ethz.ch

**Kevin Roth**[*]
Dept of Computer Science
ETH Zürich
kevin.roth@inf.ethz.ch

**Gregor Bachmann**[*]
Dept of Computer Science
ETH Zürich
gregor.bachmann@inf.ethz.ch

**Sebastian Nowozin**
Microsoft Research
Cambridge, UK
Sebastian.Nowozin@microsoft.com

**Thomas Hofmann**
Dept of Computer Science
ETH Zürich
thomas.hofmann@inf.ethz.ch

## Abstract

The "*cold posterior effect*" (CPE) in Bayesian deep learning describes the uncomforting observation that the predictive performance of Bayesian neural networks can be significantly improved if the Bayes posterior is artificially sharpened using a temperature parameter $T < 1$. The CPE is problematic in theory and practice and since the effect was identified many researchers have proposed hypotheses to explain the phenomenon. However, despite this intensive research effort the effect remains poorly understood. In this work we provide novel and nuanced evidence relevant to existing explanations for the cold posterior effect, disentangling three hypotheses: 1. The *dataset curation hypothesis* of Aitchison (2020): we show empirically that the CPE does not arise in a real curated data set but can be produced in a controlled experiment with varying curation strength. 2. The *data augmentation hypothesis* of Izmailov et al. (2021) and Fortuin et al. (2021): we show empirically that data augmentation is sufficient but not necessary for the CPE to be present. 3. The *bad prior hypothesis* of Wenzel et al. (2020): we use a simple experiment evaluating the relative importance of the prior and the likelihood, strongly linking the CPE to the prior. Our results demonstrate how the CPE can arise in isolation from synthetic curation, data augmentation, and bad priors. Cold posteriors observed "in the wild" are therefore unlikely to arise from a single simple cause; as a result, we do not expect a simple "fix" for cold posteriors.

## 1 Introduction

Deep neural networks have achieved great success in predictive accuracy for supervised learning tasks. Unfortunately, however, they still fall short in giving useful estimates of their predictive *uncertainty*, i.e. meaningful confidence values for how certain the model is about its predictions (Ovadia et al., 2019). Quantifying uncertainty is especially crucial in real-world settings, which often involve data distributions that are shifted from the one seen during training (Quionero-Candela et al., 2009).

Bayesian deep learning combines deep learning with Bayesian probability theory. *Bayesian neural networks* (BNNs) learn a distribution over model parameters or equivalently sample an ensemble of likely models given the data, promising better generalization performance and principled uncertainty quantification (robust predictions) (Neal, 1995; MacKay, 1992; Dayan et al., 1995).

35th Conference on Neural Information Processing Systems (NeurIPS 2021).

In Bayesian deep learning we either learn a distribution $q(\boldsymbol{\theta})$ over models compatible with the data, i.e. $q(\boldsymbol{\theta}) \simeq p(\boldsymbol{\theta}|\mathcal{D})$, or we *sample* an ensemble of models $\boldsymbol{\theta}_1, \ldots, \boldsymbol{\theta}_K \sim p(\boldsymbol{\theta}|\mathcal{D})$ from the posterior over likely models

$$p(\boldsymbol{\theta}|\mathcal{D}) \propto p(\mathcal{D}|\boldsymbol{\theta})\, p(\boldsymbol{\theta}), \tag{1}$$

where in the i.i.d. setting $p(\mathcal{D}|\boldsymbol{\theta}) = \prod_{i=1}^{n} p(y_i|\mathbf{x}_i, \boldsymbol{\theta})$ is the likelihood, relating the model we want to learn to the observations $\mathcal{D} = \{(\mathbf{x}_i, y_i)\}_{i=1}^{n}$, and $p(\boldsymbol{\theta})$ is a proper prior, e.g. a Gaussian density.

BNN predictions involve model averaging: rather than betting everything on a single point estimate of the parameters, we predict on a new instance $\mathbf{x}$ by *averaging* over all likely models compatible with the data,

$$p(y|\mathbf{x}, \mathcal{D}) = \int p(y|\mathbf{x}, \boldsymbol{\theta})\, p(\boldsymbol{\theta}|\mathcal{D})\, \mathrm{d}\boldsymbol{\theta} \quad \simeq \quad \sum_{k=1}^{K} p(y|\mathbf{x}, \boldsymbol{\theta}_k), \ \text{ where } \boldsymbol{\theta}_k \sim p(\boldsymbol{\theta}|\mathcal{D}). \tag{2}$$

Equation 2 is also known as the *posterior predictive* or *Bayesian model average*. Note that, in practice, solving the integral exactly is impossible. However, we can approximate it via Monte Carlo sampling using an ensemble of models $\boldsymbol{\theta}_k \sim p(\boldsymbol{\theta}|\mathcal{D})$, see Section C in the Appendix for further details.

The two main inference paradigms to learn a distribution over model parameters $q(\boldsymbol{\theta}) \simeq p(\boldsymbol{\theta}|\mathcal{D})$ respectively to sample from the posterior $\boldsymbol{\theta}_1, \ldots, \boldsymbol{\theta}_K \sim p(\boldsymbol{\theta}|\mathcal{D})$ are Variational Bayes (VB) (Hinton and Van Camp, 1993; MacKay et al., 1995; Barber and Bishop, 1998; Blundell et al., 2015) and Markov Chain Monte Carlo (MCMC) (Neal, 1995; Welling and Teh, 2011; Chen et al., 2014; Ma et al., 2015). We will focus on MCMC methods as they are simple to implement and can be scaled to large models and datasets when used with stochastic minibatch gradients (SG-MCMC) (Welling and Teh, 2011; Chen et al., 2014; Li et al., 2016).

In recent years, the Bayesian deep learning community has developed increasingly accurate and efficient approximate inference procedures for deep BNNs (cf. references in the paragraph above). Despite this algorithmic progress, however, important questions surrounding BNNs remain unanswered to this day. A recent and particularly prominent one concerns the "*cold posterior effect*" (CPE), which describes the observation that the predictive performance of BNNs can be significantly improved if the Bayes posterior is artificially sharpened $p(\boldsymbol{\theta}|\mathcal{D})^{1/T}$ using a temperature parameter $T < 1$ (Wenzel et al., 2020). Such cold posteriors sharply deviate from the Bayesian paradigm but are commonly used as heuristics in practice, see Section 2.3 in (Wenzel et al., 2020).

The CPE is problematic in theory and practice, and since the effect was identified many researchers have proposed hypotheses to explain the phenomenon. There has been an ongoing debate questioning the roles of isotropic Gaussian priors (Wenzel et al., 2020; Zeno et al., 2020; Fortuin et al., 2021), the likelihood model (Aitchison, 2020), inaccurate inference (Izmailov et al., 2021; Wenzel et al., 2020), and data augmentation (Izmailov et al., 2021; Fortuin et al., 2021). However, despite this intensive research effort the effect remains poorly understood.

**Contributions:** We provide novel and nuanced evidence relevant to existing explanations for the cold posterior effect, disentangling the roles of curation, data augmentation and the prior:

- The *dataset curation hypothesis* of Aitchison (2020): we show empirically that the CPE does not arise in a real curated data set but can be produced in a controlled experiment with varying curation strength.

- The *data augmentation hypothesis* of Izmailov et al. (2021) and Fortuin et al. (2021): we show empirically that data augmentation is sufficient but not necessary for the CPE to be present.

- The *bad prior hypothesis* of Wenzel et al. (2020): we use a simple experiment evaluating the relative importance of the prior and the likelihood, strongly linking the CPE to the prior.

Our results demonstrate how the CPE can arise in isolation from synthetic curation, data augmentation, and bad priors. In fact, we are able to reproduce the cold posterior effect with each of the three factors alone. Cold posteriors observed "in the wild" are therefore unlikely to arise from a single cause; as a result, we do not expect a simple "fix" for cold posteriors.

## 2 Cold Posteriors: Background & Related Work

The "*cold posterior effect*" (CPE) states that among all temperized posteriors $p(\boldsymbol{\theta}|\mathcal{D})^{1/T}$ the best posterior predictive performance on holdout data is achieved at temperature $T < 1$ (Wenzel et al., 2020). Formally, tempering the posterior corresponds to a $1/T$-scaling of the potential energy function $U(\boldsymbol{\theta})$,

$$p(\boldsymbol{\theta}|\mathcal{D})^{1/T} \propto \exp\left(-\frac{1}{T}U(\boldsymbol{\theta})\right), \quad \text{where } U(\boldsymbol{\theta}) = -\sum_{i=1}^{n} \log p(y_i|\boldsymbol{x}_i, \boldsymbol{\theta}) - \log p(\boldsymbol{\theta}), \quad (3)$$

i.e., both the log-likelihood and the log-prior are scaled by $1/T$. For $T = 1$ we have the Bayes posterior, whereas for $T \to 0$ we obtain a sequence of distributions which have their mass more and more confined around the MAP mode of the distribution (Leimkuhler et al., 2019). We can thus think of the $T \to 0$ limit of posterior inference as MAP estimation.

Note that, besides tempering the posterior as in Equation 3, one can also temper only the likelihood resp. scale only the log-likelihood, as is commonly done in VB, see Section 2.3 in (Wenzel et al., 2020). It is worth pointing out though that both variants are practically equivalent if the prior variance is multiplicative in log-prior pre-factors (such as for Gaussian priors) and if sufficiently many prior variances are grid-searched over as part of the inference pipeline, such that for the best performing posterior-tempered model there is a corresponding likelihood-tempered model with variance scaled by $1/T$ in the grid and vice versa.

There are three main building blocks in Bayesian deep learning that may be at fault for the CPE to emerge: (i) *model misspecification*: the likelihood model could be misspecified (Wenzel et al., 2020; Adlam et al., 2020; Aitchison, 2020; Zeno et al., 2020), (ii) *bad priors*: the priors currently used in deep BNNs may be inadequate (Wenzel et al., 2020; Fortuin et al., 2021), or (iii) *inaccurate inference*: the inference method might not yield an accurate enough approximation to the true posterior (Wenzel et al., 2020; Adlam et al., 2020; Fortuin et al., 2021; Izmailov et al., 2021). Next we review some of the most prominent hypotheses for the emergence of the CPE.

**Inaccurate Inference Hypothesis:** While recent works on the CPE have taken great care to ensure that their SG-MCMC based inference procedure yields as accurate an approximation to the true posterior as possible, it remains difficult to definitively assess the approximation accuracy without having access to the (intractable) true posterior. To rule out obvious problems with the inference mechanism, Wenzel et al. (2020) proposed a set of diagnostics, based on comparing ensemble statistics to their theoretically known values. We have closely monitored these diagnostics in our experiments, however, despite our own extensive efforts to ensure accurate inference, we cannot exclude the possibility that our inference may be inaccurate even though the diagnostics match.

To investigate the inference hypothesis further, together with other foundational questions in Bayesian deep learning, Izmailov et al. (2021) recently applied full-batch Hamiltonian Monte Carlo (HMC)[1], which is considered to be the gold-standard in terms of inference accuracy, to the models of Wenzel et al. (2020), showing that with HMC inference and when data augmentation is turned off the CPE disappears. While these results can easily be misconstrued as evidence that the approximate inference is at fault in the CPE, there are good reasons to think otherwise.

For instance, Izmailov et al. (2021) show, using the code of Wenzel et al. (2020), that turning off data augmentation alone is sufficient to remove the CPE (cf. Table 7 in Izmailov et al. (2021) Appendix G). We too can confirm that the CPE does not arise with SG-MCMC based inference applied to Wenzel et al. (2020)'s models when data augmentation is turned off, cf. Figure 1 in Section 4.

From this we can already conclude that either SG-MCMC inference is accurate enough for this specific setting, or inaccurate inference is not necessary for the CPE to emerge[2].

A more direct counter argument follows from (Adlam et al., 2020), which have demonstrated that there can be a CPE in Gaussian Processes (GP) regression, where the posterior has a closed form solution, provided that the aleatoric uncertainty is overestimated. Hence, the CPE can arise in a

---

[1]Izmailov et al. (2021) parallelized the HMC computation over *hundreds* of Tensor Processing Units (TPUs)

[2]We can also conclude that data augmentation is sufficient for the CPE to arise (although it is not necessary, as we will see later).

setting where exact inference is possible. They also provide experimental evidence that the CPE can arise in classification tasks in the infinite-width neural network Gaussian process (NNGP) limit. Thus, while inaccurate inference may be sufficient, it does not appear to be necessary for the CPE to emerge. Note that a similar observation was made in (Grünwald, 2012; Grünwald et al., 2017), which demonstrated benefits of tempering, with $T > 1$ in their setting, in the context of exact inference.

**Curation Hypothesis (*model misspecification*):** Aitchison (2020) devises a theory that attributes the effectiveness of tempering to the fact that standard benchmark datasets such as CIFAR-10 are carefully curated and that we should take this curation into account by tempering BNNs. In Aitchison (2020)'s model of curation (actual dataset curation may differ), a datapoint $x$ is added to the dataset if and only if *all* $S$ labellers independently agree on the label $y_s$ to be assigned to $x$, while the datapoint is discarded if at least one pair of labellers $s, s'$ disagrees $y_s \neq y_{s'}$. The main argument put forward by Aitchison (2020) is that if we a priori know that the dataset is curated in the sense described above, we should take this into account in our likelihood model. The proposed likelihood to be used in the case of curation should then be of the following form

$$p(y, \boldsymbol{x}|\boldsymbol{\theta}) \propto p(\{y_s = y\}_{s=1}^{S}|\boldsymbol{x}, \boldsymbol{\theta}) = \prod_s p(y_s = y|\boldsymbol{x}, \boldsymbol{\theta}) . \tag{4}$$

Thus, assuming that the labellers are i.i.d., the probability of consensus on label $y$ is

$$p(y, \boldsymbol{x}|\boldsymbol{\theta}) \propto p(y|\boldsymbol{x}, \boldsymbol{\theta})^S , \tag{5}$$

which corresponds to a cold posterior where only the log-likelihood is re-scaled while the prior is not (cf. discussion at the beginning of this Section). Aitchison (2020) argues that we should observe $S \approx 1/T$, i.e., the optimal temperature should roughly[3] be inversely proportional to the total number of labellers involved in the curation of a datapoint.

Note that Aitchison (2020)'s model of curation only includes a data point if *all* labellers agree, but in practice, datasets are often collected with some tolerance of labeller disagreement, e.g. in that a datapoint is included if a certain fraction of labellers agree. However, a more realistic (weaker) model of curation, that filters out fewer "hard" instances from the pool of uncurated datapoints, would only give rise to a weaker, less pronounced CPE: if we do not observe the CPE for Aitchison (2020)'s simplistic model of curation, which is the most extreme form of curation imaginable, we would not expect to observe it under more realistic models of curation either. See Section B in the Appendix for additional details on how popular datasets like CIFAR-10 or SVHN were collected.

Finally, we would like to point out that Adlam et al. (2020) made a similar, albeit somewhat more general argument regarding curation resp. mismatch of aleatoric uncertainty. Specifically, they show that the CPE can arise if the model overestimates the aleatoric uncertainty, which is naturally reduced when the dataset is curated.

**Data Augmentation Hypothesis (*model misspecification*):** Current deep learning practices use a number of techniques, including data augmentation, that technically do not obey the likelihood principle (see Appendix K in (Wenzel et al., 2020) for an in-depth discussion of so-called "dirty likelihoods"). The data augmentation hypothesis specifically says that the CPE is largely an artifact of using data augmentation and that turning off data augmentation is sufficient to remove the CPE.

The hypothesis has recently gained traction with both Izmailov et al. (2021) and Fortuin et al. (2021) pointing out that turning off data augmentation is sufficient to remove the CPE in Wenzel et al. (2020)'s ResNet CIFAR10 setting (cf. Table 7 in Izmailov et al. (2021) Appendix G and Figure A.11 in Fortuin et al. (2021) Appendix A). On the other hand, Wenzel et al. (2020)'s CNN-LSTM IMDB model already had a clean likelihood function and still gave rise to a CPE. From these observations we can already conclude that data augmentation is sufficient but not necessary for the CPE to arise.

As the performance of deep neural networks is often significantly better when using some form of data augmentation, it is not really an option to just turn it off in BNNs, while properly accounting for it in Bayesian inference does not seem trivial either. On the one hand, data augmentation affects the data points that enter the likelihood function. However, while data augmentation may increase the

---

[3]To obtain an exact correspondence, we would need access to the discarded datapoints in order to be able to marginalize them out, cf. "marginalise over unknown latents" in Section 3 in (Aitchison, 2020) for how to account for the discarded images in a proper Bayesian sense

amount of data seen by the model, that increase is certainly not equal to the number of times each data point is augmented (after all, augmented data is not independent from the original data). On the other hand, considering data augmentation as a form of regularization (constraining the classification functions to be invariant to certain transformations), one can argue that it should be represented in the prior (Wilk et al., 2018). It remains an interesting open problem how to properly account for data augmentation in a Bayesian sense.

**Bad Prior Hypothesis:** Isotropic Gaussian priors are the de facto standard for modern Bayesian neural network inference (Fortuin et al., 2021). However, it is questionable whether such simplistic priors are optimal and whether they accurately reflect our true beliefs about the weight distributions. The bad prior hypothesis says that the CPE may only be an epi-phenomenon of a misspecified prior. The underlying argument is as follows: in classic Bayesian learning the number of parameters remains small and the prior is quickly dominated by the data. In contrast, in Bayesian deep learning the model dimensionality is typically on the same order if not larger than the dataset size (Kaplan et al., 2020). For such large models the prior will not be dominated by the data and will continue to exert an influence on the posterior. Hence the prior is critical.

The hypothesis has already been put forward by Wenzel et al. (2020), who reported that the CPE becomes stronger with increasing model dimensionality. It recently got additional empirical support by Fortuin et al. (2021), who found that the CPE can be partially alleviated by using heavy-tailed non-Gaussian priors. More specifically, they find that for fully connected neural networks (FCNNs), heavy-tailed priors can both improve predictive performance and alleviate the CPE. For convolutional neural networks (CNNs), the CPE can also be removed with heavy-tailed priors, however, the resulting performance gains are less striking. On the other hand, the performance of CNNs can be improved with correlated priors, although they no longer appear to alleviate the CPE.

Finally, we note that the prior that ultimately matters is the prior over functions that is induced when a prior over parameters is combined with the functional form of a neural network architecture (Wilson and Izmailov, 2020; Izmailov et al., 2021). Still, this does not render the prior over parameters irrelevant, as innocent-looking priors may inadvertently be highly informative, for instance placing large prior mass on undesirable functions.

## 3   Testing the Relative Influence of the Prior

A straightforward way to assess the bad prior hypothesis is to monitor the CPE while continuously trading off the relative influence between the prior term and the likelihood term: if the CPE becomes stronger as the relative influence of the prior increases, this would be an indication that the prior is poor. The relative weight of the prior versus the likelihood in Bayesian inference is given by a simple factor: the dataset size $n$. To see this, recall the posterior energy function in Equation 3. Note how the log-likelihood is a sum over $n$ datapoints, i.e. it scales with the dataset size $n$, while the log-prior is independent of $n$. This means that the relative influence of the log-prior vanishes at a rate of $1/n$ compared to the influence of the log-likelihood. In other words, the prior will exert its strongest influence for relatively small dataset sizes $n$.

In order to test the relative influence of the prior, we devise a simple experiment in which we train BNNs on random sub-samples of different sizes, recording the optimal temperature for each value of the dataset size $n$. For smaller $n$ the relative importance of the prior with respect to the likelihood is larger, while for larger $n$ the prior has smaller influence on the posterior. By varying the dataset size $n$ we can test the following two hypotheses:

- If a bad prior causes the CPE, we would expect to see a stronger CPE for smaller dataset sizes.
- From the theory of curation (Aitchison, 2020) we would expect that random subsamples of the dataset do not cause a change in the optimal temperature $T^*$, i.e. we would expect the same $T^*$ regardless of the dataset size $n$.

Note that when performing the sub-sampling experiments, care must be taken to ensure that the SG-MCMC inference has the same total number of parameter gradient updates across all data set sizes. In particular, fixing the batch size, we have to increase the cycle length (i.e. the number of epochs per cycle) for smaller datasets. This ensures that the number of samples from the posterior and the overall number of gradient updates are the same across dataset sizes. Note also that we temper

the whole posterior (not just the likelihood) to keep the relative influence of the log-likelihood and log-prior terms the same, cf. discussion at the beginning of Section 2.

Finally, we note that a similar sub-sampling experiment to evaluate different priors was suggested in (Atanov et al., 2018). Although the proposed experiment is a rather straightforward method to test the quality of a prior, recent works that study the CPE (Adlam et al., 2020; Wilson, 2019; Aitchison, 2020; Zeno et al., 2020; Fortuin et al., 2021; Izmailov et al., 2021) did not perform such analysis, despite the ongoing debate on the role of the likelihood and the prior in the CPE.

## 4 CPE: A Symptom with Many Causes?

We use the SG-MCMC implementation of Wenzel et al. (2020) for all our experiments[4]. In particular, we adopt a cyclical step size schedule and, optionally, layerwise preconditioning. We explicitly specify when critical features - e.g. data augmentation - are adopted, otherwise we refer to Appendix D for a detailed description of the experimental setup, including the hyperparameters and estimates of the compute resources we used. For each experiment, we consider six temperature parameters $T \in [10^{-3}, 1]$, where a separate Markov chain was used for each temperature. We generally report performance in terms of the test cross-entropy, whereas the corresponding accuracy and uncertainty measurements can be found in the Appendix. Shaded areas in the plots below denote standard errors w.r.t. the number of random seeds (three in our case). We also define the *CPE-ratio* "CPER",

$$\text{CPER} = \ell_{T^*}/\ell_{T=1} \in (0, 1],\tag{6}$$

as the ratio between the cross-entropy loss at the optimal temperature $T^*$ versus $T = 1$ (Bayes posterior). A low CPER indicates that the performance of the tempered posterior is significantly better than the Bayes posterior. We perform experiments on SVHN (Netzer et al., 2011) and CIFAR-10 (Krizhevsky and Hinton, 2009), both of which are curated (see Appendix B). We use ResNet-20 neural networks (He et al., 2016) with Gaussian priors $\mathcal{N}(0, 1)$, unless stated otherwise. Further results, including for CNN-LSTM on IMDB and MLP on MNIST, can be found in the Appendix.

**Starting point: no CPE without data augmentation** We run SG-MCMC on SVHN and CIFAR-10 without data augmentation. As can be seen in Figure 1, we observe that SG-MCMC inference on the full dataset $\mathcal{D}$ does not show any sign of a CPE, i.e. $T = 1$ is close to optimal. From this we can conclude that *either SG-MCMC inference is accurate enough for this specific setting, or inaccurate inference is not necessary for the CPE to emerge*. We can also conclude that *the curation of SVHN and CIFAR10 does not give rise to a CPE*, which is somewhat surprising from Aitchison (2020)'s consensus theory standpoint. The curation hypothesis is discussed in more detail below.

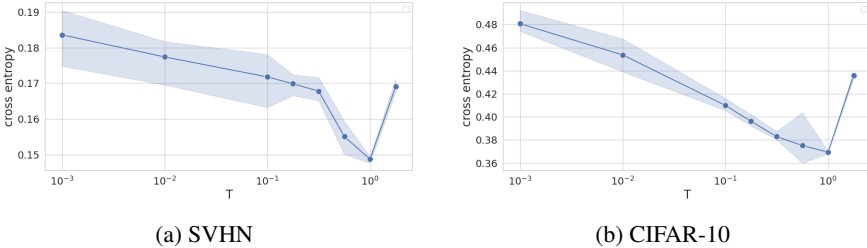

(a) SVHN          (b) CIFAR-10

Figure 1: Test cross-entropy as a function of temperature $T$ for (a) SVHN and (b) CIFAR-10. We can see that $T = 1$ is optimal for SVHN and CIFAR-10.

### 4.1 CPE with curation, data augmentation, and random sub-sampling

**Summary**: The CPE can arise in case of: synthetic curation, i.e. when the the role of the labellers is played by trained neural networks, data augmentation and random sub-sampling.

---

[4]https://github.com/google-research/google-research/tree/master/cold_posterior_bnn

**CPE and synthetic curation** We now test whether curation can cause the CPE in a simulated environment in which we control the number of labellers and can scale the amount of curation to large levels. We do so by performing synthetic curation as follows:

- We first split $\mathcal{D}$ into two non-intersecting sets: a pre-training dataset $\mathcal{D}_{pre}$ and a dataset $\mathcal{D}_{tr}$.

- We train a probabilistic classifier $\hat{\mathcal{S}}$ on $\mathcal{D}_{pre}$, that learns a categorical distribution over the labels (given by the output of the softmax activation of a neural network).

- We use $S$ copies drawn from $\hat{\mathcal{S}}$ to independently re-label $\mathcal{D}_{tr}$, effectively simulating the behavior of $S$ i.i.d. labellers. The labels are obtained by sampling from the categorical distribution learned by the network. The more uncertain the model is about a prediction for some input, the more disagreement between labellers we expect for that input.

- Using the labels induced by $\hat{\mathcal{S}}$, we apply the curation procedure described in (Aitchison, 2020) and summarized earlier in Section 2, to filter $\mathcal{D}_{tr}$ and obtain $\mathcal{D}_{tr}^{cur}$. Note that the consensus label does not necessarily match with the original label (which can happen for images where the model is confident on the wrong label).

As the labeller classifier $\hat{\mathcal{S}}$, we train a ResNet on $\mathcal{D}_{pre}$ with the Adam optimizer. Details of the training procedure can be found in Section D.3 in the Appendix.

We perform SG-MCMC inference on the curated dataset $\mathcal{D}_{tr}^{cur}$ using $S$ labellers, for various values of $S$. The cross-entropy is evaluated both on the original test set $\mathcal{D}_{test}$ - which is simply re-labeled according to the trained model $\hat{\mathcal{S}}$ - and the curated one $\mathcal{D}_{test}^{cur}$ using the same number of $S$ labellers as in the training set. Surprisingly, *we do not observe any cold posterior effect when only the training set is curated*, as shown in Figure 2a. This confirms results of (Aitchison, 2020) (Figure 4D). However, as shown in Figure 2b, *curation of both the training and test set causes a CPE*.

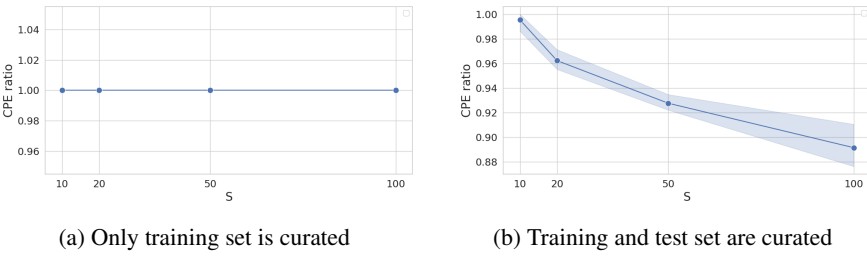

(a) Only training set is curated     (b) Training and test set are curated

Figure 2: CPER as a function of the number of labellers $S$. (a) there is no CPE when only the training set is curated. (b) The CPE arises when the test set is curated, too.
.

**CPE and data augmentation** Recent works (Izmailov et al., 2021; Fortuin et al., 2021), have identified data augmentation as a cause for the CPE. We confirm that data augmentation can cause the CPE on SVHN and CIFAR-10, as shown in Figure 3. In the next paragraph, we provide evidence that the CPE can arise even without data augmentation. From these observations we can conclude that *data augmentation is sufficient but not necessary for the CPE to arise*. Note that data augmentation could hint at a problem with the prior, too (when considering data augmentation as a form of regularization). Details on the kind of data augmentation used for each data set can be found in Section E.4.

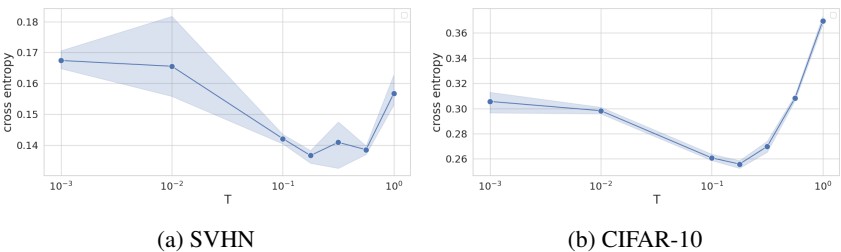

(a) SVHN     (b) CIFAR-10

Figure 3: CPE on (a) SVHN and (b) CIFAR-10, both with data augmentation.

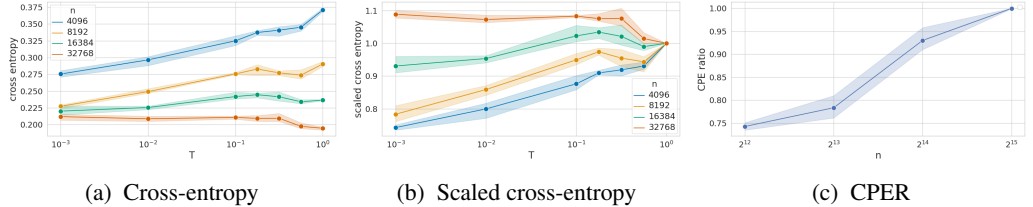

| (a) Cross-entropy | (b) Scaled cross-entropy | (c) CPER |
|---|---|---|

Figure 4: SVHN sub-sampling experiment. (a) BNN performance in terms of test cross-entropy for different temperature values. Each line represents a different subsample size, from $n = 2^{12} = 4096$ to $2^{15} = 32768$. (b) same plot, but the test cross-entropy values are scaled by the value at $T = 1$. (c) CPER metric as a function of $n$. Note how the CPE is stronger for smaller $n$.

**CPE and random sub-sampling**   We now discuss the random sub-sampling experiment to investigate the quality of the prior. We run SG-MCMC *without data augmentation* on subsets of SVHN with different sample sizes $n$. As explained in Section 3, we ensure that the SG-MCMC inference has the same total number of parameter gradient updates across all data set sizes[5]. The results in Figure 4 show that sub-sampling alone can cause CPE. Note how the CPE is stronger for smaller $n$, as is evident from the CPE ratio in Figure 4c. As the influence of the prior is larger on smaller datasets, *this is a strong indication that the prior is at fault*[6].

### 4.2   Comparing the relative influence of curation, data augmentation and sub-sampling

In the previous paragraph, we have seen that sub-sampling alone can cause the CPE to arise. Here we investigate the effect of sub-sampling on top of curation and data augmentation resp. the additional impact of curation and data augmentation on a sub-sampled dataset. The precise description of the experimental setup can be found in Section D.4 in the Appendix. The results are shown in Figure 5. The plot can be read in two ways: either one looks at a fixed dataset size $n$ and compares the impact of curation (blue) and data augmentation (green) over the original test set (orange), or one looks at the relative change in CPER induced by sub-sampling for a given curve (e.g. how much each curve drops when going from $n = 16384$ to $n = 8192$).

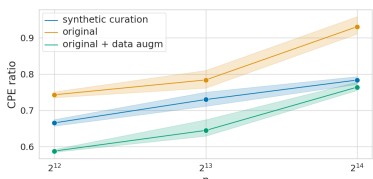

Figure 5: Comparing the CPER for random sub-sampling with curation and data augmentation on datasets of size 4096, 8192 and 16384.

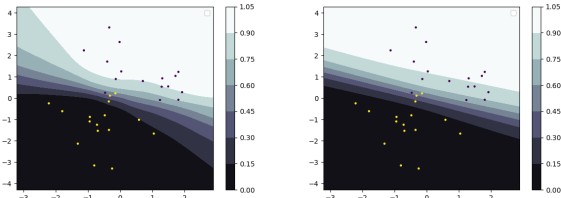

Figure 6: Decision boundary of 1 hidden layer MLP on toy dataset of size $n = 32$, (left) $T = 1$, (right) $T = 10^{-2}$. Color shading according to posterior mean prediction.

### 4.3   Do standard Gaussian priors give too much weight to complicated hypotheses?

**Summary**: The CPE can arise even in a small scale experiment in which a linear separator is optimal. An analysis of the decision boundary suggests that the sharpened posterior obtained through tempered MCMC induces simpler functions than the Bayes posterior at $T = 1$.

---

[5]We also repeat the sub-sampling experiment in an ablation setting where the number of gradient steps decreases for smaller datasets by reducing the number of epochs per cycle. The results are almost indistinguishable, as can be seen in Section E in the Appendix.

[6]Note that issues with the prior could also underlie the data augmentation induced CPE observed above: It is difficult to isolate the effect of data augmentation without being able to exclude potential issues with the prior, i.e. without knowing for sure what a good prior for BNNs is.

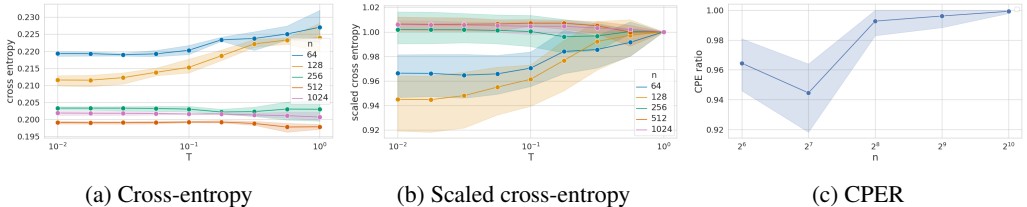

| (a) Cross-entropy | (b) Scaled cross-entropy | (c) CPER |

Figure 7: Testing the relative influence of the prior on a toy dataset. (a) Cross-entropy (CE) as a function of temperature $T$, (b) same plot, but the test cross-entropy values are scaled by the value at $T = 1$, (c) CPE ratio as a function of dataset size $n$. We can see that the CPE becomes stronger for smaller dataset size $n$.

Here we investigate the influence of the prior in a synthetic toy dataset. To this end, we generate two clusters of datapoints from two 2D Gaussians with variance $\sigma^2 = 1$ centered at $(-1, -1)$ and $(1, 1)$ respectively, in which the Bayes optimal classifier is given by the straight line $y = -x$.

To investigate the influence of the prior, we run full-batch MCMC for a 1 hidden layer MLP on subsets of the toy dataset with varying sample sizes $n$ and record the CPE across various temperatures $T$. Note that the prior is well-specified in the sense that there is a parameter setting that achieves perfect classification (clearly a 1 hidden layer MLP can learn the optimal separator). The results, in Figure 7, show that the CPE becomes stronger for smaller dataset size $n$.

An analysis of the decision boundary, shown in Figure 6, suggests that the sharpened posterior obtained through tempered MCMC induces simpler functions than the Bayes posterior at $T = 1$. We consider it highly relevant future work to investigate if this holds also for real world problems

## 5 Discussion & Conclusion

The "*cold posterior effect*" (CPE) in Bayesian deep learning describes the disturbing observation that the predictive performance of Bayesian neural networks (BNNs) can be significantly improved if the Bayes posterior is artificially sharpened using a temperature parameter $T < 1$. Since the CPE was identified many researchers have proposed hypotheses to explain the phenomenon. However, despite this intensive research effort the effect remains poorly understood. We have provided novel and nuanced evidence relevant to existing explanations for the CPE.

Our results demonstrate how the CPE can arise in isolation from synthetic curation, data augmentation, and random sub-sampling. Specifically, we have confirmed that there is no CPE on SVHN and CIFAR-10 if data augmentation is turned off (Figure 1), which is somewhat surprising from the curation theory standpoint (since both datasets are curated). On the other hand, we have shown that the CPE can arise when both the training and test set are synthetically curated, i.e. when the role of the labellers is played by trained neural networks (Figure 2). Most importantly, we have shown that the CPE can also arise when a dataset that does not show any sign of CPE is randomly sub-sampled (Figure 4), providing a strong indication that the prior is at fault in the CPE (the relative influence of the prior over the likelihood increases with decreasing dataset size).

The CPE may be a symptom with many causes. Since many of the recent deep learning advances, such as data augmentation, batch normalization and initialization distributions have been designed specifically for DNNs, it is not surprising that tempering, which gets BNNs closer to DNNs, improves their performance. The implication of this is that the Bayesian deep learning community has to find their own "advances" specifically tailored to BNNs.

Another conclusion of our work is that priors *do* matter in Bayesian deep learning. In classic Bayesian learning the number of parameters remains small and the prior is quickly dominated by the data. In contrast, in Bayesian deep learning the model dimensionality is typically on the same order if not larger than the dataset size. For such large models the prior will not be dominated by the data and will continue to exert an influence on the posterior. Moreover, issues with the prior could also underlie some of the other hypotheses, e.g. to account for data augmentation, which can be considered a form of regularization, we might need to represent it in the prior. We therefore consider it to be timely to study suitable priors for deep BNNs.

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
