# A  Further Related Work

**"Warm" Posteriors:**  Motivated by the behavior of Bayesian inference in *misspecified* models Grün-wald et al. (2017); Jansen (2013) extensively studied the so called "generalized" Bayesian inference, i.e, the Bayes posterior in which only the likelihood is tempered. In particular, the "Safe Bayes" framework (Grünwald, 2012, 2011; Grünwald et al., 2017) was developed to tune the temperature parameter. However, these works consider only "warm posteriors" $T > 1$ (the inverse temperature is called "learning rate" in the relevant literature), as a way to learn under model misspecification. Warm posteriors can arise in a context where the model is misspecified, for instance by assuming homoscedastic noise where the data-generating noise is heteroscedastic. Under this misspecification, the model overfits the datapoints, despite the fact that the prior is well-specified (for instance in their case it is centered around the best performing and non-overfitting solution). We hypothesize that in Grünwald et al. (2017) the prior favours simple models, hence it is beneficial to put more weight onto the prior and use warm posterior. The opposite might happen in BNNs: cold posteriors counteract the effect of a bad prior that tends to prefer overcomplicated solutions. Finally, we mention the work of Bhattacharya et al. (2019), in which the authors develop *fractional posteriors* with the goal of decreasing posterior concentration.

# B  Dataset Collection & Curation

Here we review the way that the two datasets that are mainly used in our experiments, SVHN and CIFAR-10, have been collected and curated.

**SVHN**  The Street View House Numbers dataset (Netzer et al., 2011), which is divided into a training corpus $\mathcal{D}$ of around 73257 training images and a test set $\mathcal{D}_{te}$ of around $26k$ images. Although we do not know the exact number of labellers, the dataset has undergone a curation procedure in the sense of Aitchison (2020). In particular, AMT was adopted[7] (quoting from Netzer et al. (2011), "The SVHN dataset was obtained from a large number of Street View images using a combination of automated algorithms and the Amazon Mechanical Turk (AMT) framework").

**CIFAR-10**  In CIFAR-10 (Krizhevsky and Hinton, 2009), labellers followed strict guidelines to ensure high quality labelling of the images. In particular, labellers were instructed that "it's worse to include one that shouldn't be included than to exclude one. False positives are worse than false negatives", and "If there is more than one object that is roughly equally prominent, reject". The reader is invited to review Appenidx C in Krizhevsky and Hinton (2009) .

Unfortunately, in the relevant papers there are no details on the specific curation process that was applied, e.g. the number of labellers per image, or whether *all* labellers have to agree on a label or only a subset of them.

# C  MCMC Inference

In this section we review the basics of (SG)-MCMC inference. The description of the implementation and adaptations for deep learning can be found in Section D.2 below.

The two main inference paradigms to learn a distribution over model parameters $q(\boldsymbol{\theta}) \simeq p(\boldsymbol{\theta}|\mathcal{D})$ respectively to sample from the posterior $\boldsymbol{\theta}_1, \ldots, \boldsymbol{\theta}_K \sim p(\boldsymbol{\theta}|\mathcal{D})$ are Variational Bayes (VB) (Hinton and Van Camp, 1993; MacKay et al., 1995; Barber and Bishop, 1998; Blundell et al., 2015) and Markov Chain Monte Carlo (MCMC) (Neal, 1995; Welling and Teh, 2011; Chen et al., 2014; Ma et al., 2015; Li et al., 2016). We will focus on MCMC methods as they are simple to implement and can be scaled to large models and datasets when used with stochastic minibatch gradients (SG-MCMC) (Welling and Teh, 2011; Chen et al., 2014).

Markov Chain Monte Carlo (MCMC) methods allow to sample an ensemble of models $\boldsymbol{\theta}_1, \ldots \boldsymbol{\theta}_K \sim p(\boldsymbol{\theta}|\mathcal{D})^{1/T}$ from the (tempered) posterior $p(\boldsymbol{\theta}|\mathcal{D})^{1/T}$, by performing a guided random walk in parameter space in which artificial noise is injected into the updates $\boldsymbol{\theta}_k \to \boldsymbol{\theta}_{k+1}$ in such a way that the ensemble distribution converges to the desired posterior $p(\boldsymbol{\theta}|\mathcal{D})^{1/T}$ (in the limit of small step

---

[7]https://www.mturk.com/

sizes and long enough run times) (Neal, 1995; Welling and Teh, 2011; Ma et al., 2015; Li et al., 2016). By injecting artificial noise, the algorithm explores the loss landscape instead of approaching a single point estimate $\widehat{\boldsymbol{\theta}}$.

**SG-MCMC**  Recent advances in stochastic inference through Markov Chain Monte Carlo (MCMC) methods have made the task of sampling from the posterior distribution of deep neural networks more efficient (Welling and Teh, 2011; Zhang et al., 2020; Wenzel et al., 2020). In particular, the usage of mini-batches gave rise to stochastic gradient MCMC methods (SG-MCMC) (Welling and Teh, 2011), which is further improved through various techniques such as momentum variables (Chen et al., 2014), preconditioning (Li et al., 2016), and cyclical stepsize (Zhang et al., 2020). All these methods perform stochastic updates in parameter space that come from the discretization of a stochastic process (Ma et al., 2015). For the purpose of exposition, here we mention SG-MCMC in its simplest form, given by SGLD (stochastic gradient Langevin dynamics), in which the updates have the form

$$\boldsymbol{\theta}_{t+1} = \boldsymbol{\theta}_t + \frac{\epsilon_t}{2} \left( \frac{n}{B} \sum_{i=1}^{B} \nabla \log p(y|\boldsymbol{x}_i, \boldsymbol{\theta}) + \nabla \log p(\boldsymbol{\theta}) \right) + \boldsymbol{\eta}_t, \tag{7}$$

where $B$ indicates the size of a mini-batch, and $\boldsymbol{\eta}_t \sim \mathcal{N}(0, \epsilon_t I)$. We define the gradient of the mini-match posterior energy as $\nabla \tilde{U}(\boldsymbol{\theta}) := \frac{n}{B} \sum_{i=1}^{B} \nabla \log p(y|\boldsymbol{x}_i, \boldsymbol{\theta}) + \nabla \log p(\boldsymbol{\theta})$. Welling and Teh (2011) show that if $\epsilon_t$ is such that:

$$\sum_{i=1}^{\infty} \epsilon_t = \infty \ , \quad \sum_{i=1}^{\infty} \epsilon_t^2 < \infty, \tag{8}$$

then convergence to a local maximum is guaranteed. We will use the SG-MCMC implementation of (Wenzel et al., 2020) throughout our experiments[8], which combines the aforementioned techniques, further discussed in Section D.2 below.

# D   Experimental Setup

The experimental details, including the SG-MCMC hyperparameters, are included in Table 1. Note that for the subsampling experiments on SVHN (Figure 4), the table entries in the last three columns, that have the number of epochs as units (i.e. burn-in period, cycle length, epochs), refer to the *full dataset size*. When subsampling is applied, the number of epochs are adjiusted such that the number of gradient steps is kept fixed. For instance, if half of the dataset is used, the number of epochs, cycle length epochs and burn-in epochs doubles. Finally, the experiments are executed on Nvidia DGX-1 GPU nodes equipped with 4 20-core Xeon E5-2698v4 processors, 512 GB of memory and 8 Nvidia V100 GPUs.

## D.1   Neural Network Architectures

For the SG-MCMC experiments, we use a 20-layer architecture with residual layers (He et al., 2016) and batch normalization. For the SG-MCMC experiments on the toy dataset, we use a single hidden layer fully connected net with 20 units and ReLU activation function. For MNIST dataset, we use a 3 hidden layers fully connected net with 20 units and ReLU activation function. For the IMDB dataset, we use CNN-LSTM architecture identical to the one used in Wenzel et al. (2020). The SG-MCMC method that we adopt is the one in Wenzel et al. (2020) and summarized in Sections D.2 and C. In particular, no preconditioning is used. The batch size is 128 across all experiments except for the toy dataset experiment, where the batch size equals the dataset size.

## D.2   Inference Method / Training Procedure

In this work, we will mainly use the inference method proposed in Wenzel et al. (2020), which adapts recent advances in optimization for deep learning and stochastic inference to SG-MCMC. See also Chen et al. (2016) for some interesting connections between SG-MCMC and stochastic optimization.

---

[8]https://github.com/google-research/google-research/tree/master/cold_posterior_bnn

Table 1: Architectural features and SG-MCMC hyper-parameters. The double horizontal line splits the experiments of the paper from those in the appendix.

| Experiment | Hyper-parameter | | | | | | |
| --- | --- | --- | --- | --- | --- | --- | --- |
| | data augm | batch norm | precond. | learning rate | burn-in | cycle length | epochs |
| SVHN & CIFAR-10 (Fig. 1) | ✗ | ✓ | ✗ | 0.1 | 150 | 50 | 1500 |
| SVHN (Fig.2) | ✓ | ✓ | ✗ | 0.1 | 100 | 25 | 500 |
| SVHN (Fig. 3) | ✓ | ✓ | ✗ | 0.1 | 150 | 50 | 1500 |
| CIFAR-10 (Fig. 3) | ✓ | ✓ | ✗ | 0.1 | 100 | 25 | 500 |
| SVHN (Fig. 4) | ✗ | ✓ | ✗ | 0.1 | 100 | 25 | 500 |
| Toy data (Fig. 7) | ✗ | ✗ | ✗ | 0.1 | 500 | 75 | 2000 |
| SVHN & CIFAR-10 (Fig. 15 and 14) | ✗ | ✓ | ✗ | 0.1 | 100 | 25 | 500 |
| MNIST (Fig. 8) | ✗ | ✗ | ✗ | 0.1 | 150 | 50 | 1500 |
| IMDB (Fig. 9) | ✗ | ✗ | ✗ | 0.1 | 150 | 50 | 1500 |
| CIFAR-10H (Fig. 24 and 23) | ✓ | ✓ | ✗ | adapted | 100 | 50 | 1000 |

**Momentum Variables**   Adding momentum to SGD is an optimization technique to accelerate gradient based optimization methods that is widely used in deep learning (Sutskever et al., 2013). Momentum variables were added to SG-MCMC methods in Chen et al. (2014), giving raise to the stochastic-gradient version of Hamiltonian dynamics (SG-HMC). SGLD can be modified as follows to include them:

$$\boldsymbol{\theta}_{t+1} = \boldsymbol{\theta}_t + \epsilon_t \boldsymbol{m}_{t+1} \tag{9}$$

$$\boldsymbol{m}_{t+1} = (1 - \epsilon_t \alpha)\boldsymbol{m}_t - \epsilon_t \nabla \tilde{U}(\boldsymbol{\theta}_t) + \sqrt{2}\boldsymbol{\eta}_t, \tag{10}$$

where $\alpha$ is the momentum weight.

**Layerwise Preconditioning**   A subset of our experiments were performed both with and without preconditioning, which did not make a big difference. The reported results are without preconditioning.

**Cyclical step size**   Cyclical step size was introduced by Zhang et al. (2020) to guarantee better exploration, given the fact that posterior exploration is somewhat limited in standard SGLD due to the fact that the learning rate $\epsilon_t$ must be small enough to avoid bias in estimation and MH acceptance/rejection steps. It consists in alternating updates with large learning rate, which allows to overshoot the local minima and therefore having a better *exploration* of the posterior landscape, and updates with very small learning rate, during which samples from the posterior are collected. All the details of the algorithm can be found in Wenzel et al. (2020), Section 3.

### D.3   Training the synthetic labellers

For the curation experiment, the role of the labeller is played by a neural network.

The experiment described in the first paragraph of Section 4.1 is designed to test whether curation can cause the CPE in a simulated environment in which we control the number of labellers and can scale the amount of curation to large levels. We do so by performing synthetic curation as follows:

- We first split $\mathcal{D}$ into two non-intersecting sets: a pre-training dataset $\mathcal{D}_{pre}$ and a dataset $\mathcal{D}_{tr}$.
- We train a probabilistic classifier $\hat{\mathcal{S}}$ on $\mathcal{D}_{pre}$, that learns a categorical distribution over the labels (given by the output of the softmax activation of a neural network).

- We use $S$ copies drawn from $\hat{\mathcal{S}}$ to independently re-label $\mathcal{D}_{tr}$, effectively simulating the behavior of $S$ i.i.d. labellers. The labels are obtained by sampling from the categorical distribution learned by the network. The more uncertain the model is about a prediction for some input, the more disagreement between labellers we expect for that input.

- Using the labels induced by $\hat{\mathcal{S}}$, we apply the curation procedure described in (Aitchison, 2020) and summarized earlier in Section 2, to filter $\mathcal{D}_{tr}$ and obtain $\mathcal{D}_{tr}^{cur}$. Note that the consensus label does not necessarily match with the original label (which can happen for images where the model is confident on the wrong label).

As the labeller classifier $\hat{\mathcal{S}}$, we train an 8-layer ResNet on $\mathcal{D}_{pre}$ with the Adam optimizer.

We perform SG-MCMC inference on the curated dataset $\mathcal{D}_{tr}^{cur}$ using $S$ labellers, for various values of $S$. The cross-entropy is evaluated both on the original test set $\mathcal{D}_{test}$ - which is simply re-labeled according to the trained model $\hat{\mathcal{S}}$ - and the curated one $\mathcal{D}_{test}^{cur}$ using the same number of $S$ labellers as in the training set. The network is optimized with the Tensorflow implementation of Adam optimizer (Kingma and Ba, 2014) using the default parameters.

### D.4 Relative influence of curation, data augmentation and subsampling

In this Section, we briefly add some additional thoughts and details regarding the setup of the experiment in Section 4.2.

If one inspects the core of curation, one may argue that as it consists of *removing* datapoints, the CPE is caused by using smaller datasets rather than the curation procedure itself. Therefore, we want to compare the added contribution to the CPE of curation (and data augmentation) with respect to random sub-sampling. While adding data augmentation to random sub-sampling is straightforward, in the case of curation it is not practical to find the exact number of labellers $S$ to match a desired dataset size. Instead, we proceed as follows: we choose the number of labellers such that the curated dataset contains around 17k-18k datapoints (the exact number may vary depending on the random seed).[9] Then we perform random sub-sampling to match the desired dataset size (16384, 8192, 4096 in our case) on the synthetically curated dataset. The only difference with the previous experiments is that we use the original labels, instead of the consensus labels. In this way, if one image $x$ belongs to both the synthetically curated dataset and the randomly sampled one, it has the same label $y$. We run SG-MCMC with the same parameters settings as in Section 4, paragraph "CPE and random sub-sampling".

## E    Further Experimental Results

### E.1    Subsampling on MNIST and IMDB

We repeat the subsampling experiment on MNIST. Results are shown in Figure 8. Note how for small sample sizes the CPER metric decreases significantly, indicating the presence of the cold posterior effect. The same can be noticed for IMDB (Figure 9).

### E.2    CIFAR-10 and SVHN

We report the values of the expected calibration error (ECE) Guo et al. (2017) and accuracy for some selected experiments. For the experiments regarding full SVHN and CIFAR-10 without data augmentation reported in Section 4, Figure 1, we report these in metrics in Figure 11 for CIFAR-10 and

We repeat the subsampling experiment on CIFAR-10. Results are in Figure 12. For other experiments on full SVHN and CIFAR-10, in which we use less number of epochs per cycle, see Figure 15 for SVHN and Figure 14 for CIFAR-10.

Finally, we additionally report accuracy and ECE for the SVHN subsampling experiments without both data augmentation and curation described in Section 4. See Figure 13.

---

[9]As we observe that the classifier $\tilde{S}$ is over confident on many datapoints, we decide to smooth its output distribution, keeping the rank of the probabilities unchanged. We artificially flatten the labelling distribution by raising the probabilities to the power of $\alpha$, where $\alpha < 1$, and then re-normalizing.

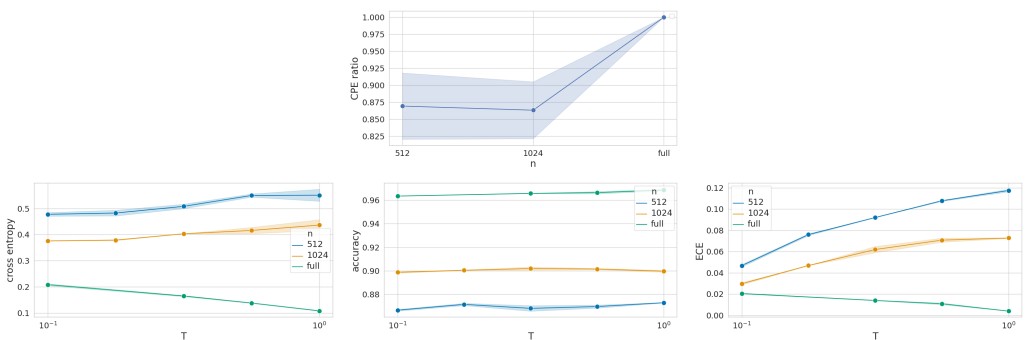

Figure 8: MNIST subsampling experiment

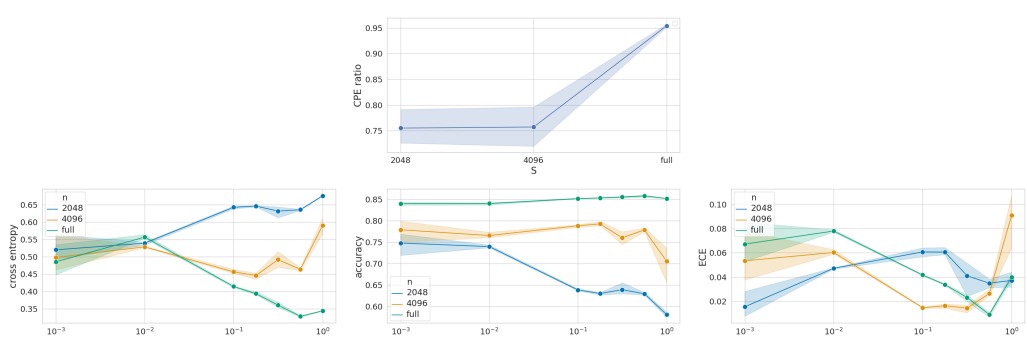

Figure 9: IMDB subsampling experiment

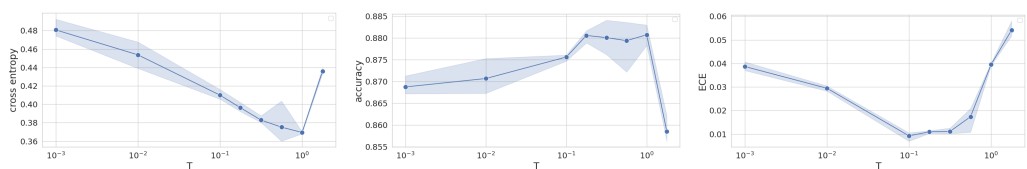

Figure 10: CIFAR full dataset experiment (no data augmentation): we additionally report accuracy and ECE.

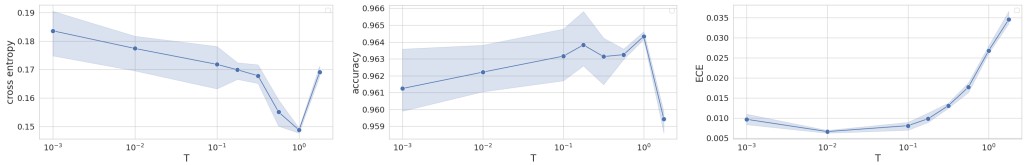

Figure 11: SVHN full dataset experiment (no data augmentation): we additionally report accuracy and ECE.

## E.3 SVHN curated

In Figure 16 and 17, we show the plots underlying the curation experiment on SVHN, summarized in Figure 2. In Figure 18 we show the underlying plots of the curation + subsampling experiment of Figure 5.

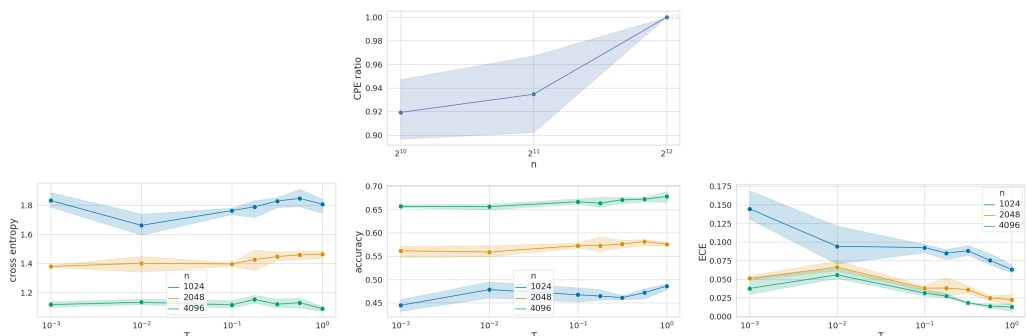

Figure 12: CIFAR subsampling experiment: we can see that even for low dataset sizes there is (almost) no CPE on CIFAR-10. In particular, performances are quite insensitive to the temperature $T$.

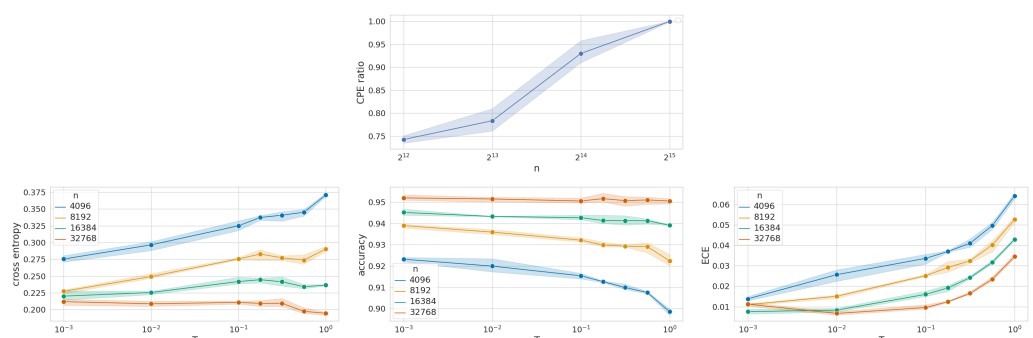

Figure 13: SVHN subsampling experiment: the plots for the CPER and cross entropy are the same as Fig. 4 and added here for completeness.

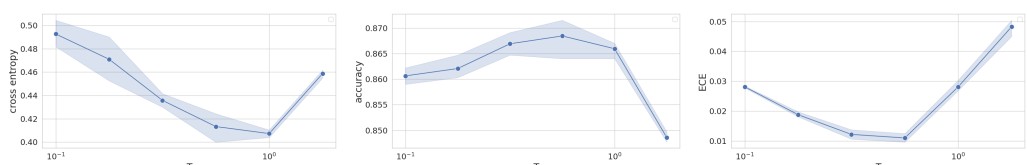

Figure 14: CIFAR-10 experiment on full dataset with smaller cycles and less samples (no data augmentation)

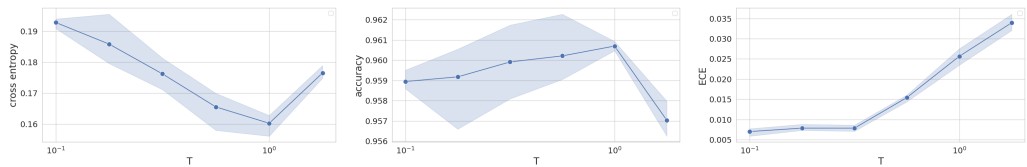

Figure 15: SVHN experiment on full dataset with smaller cycles and less samples (no data augmentation)

### E.4 Data Augmentation

When data augmentation is used, we perform a random sequence of transformations to every batch during training that causes small class-preserving changes in the images. These transformation, applied at every batch and every epoch, are as follows.

On CIFAR-10 (same as Wenzel et al. (2020)):

- random left/right flip of the image .

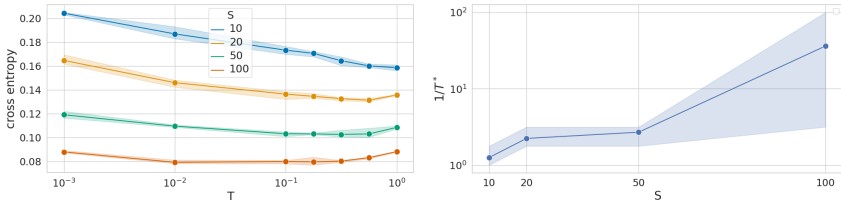

Figure 16: Plots underlying experiments when both train and test set are curated

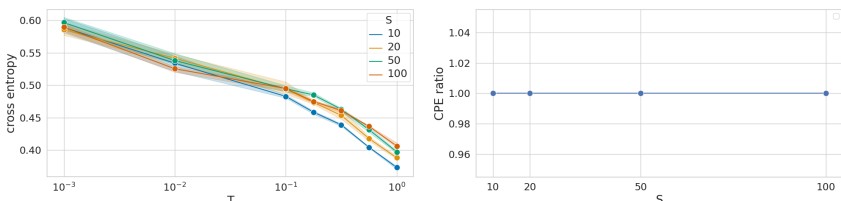

Figure 17: Plots underlying experiments when only the training set is curated: they all perform quite similarly.

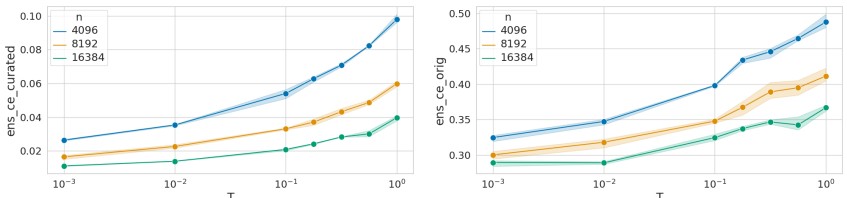

Figure 18: Curation + subsample: underlying plots. Left: curated test set. Right: original test set

- zero padding that expands the size of the input image by four pixels horizontally and vertically, and then random cropping to the original size.

On SVHN:

- adjust the contrast by a random contrast factor between 0.45 and 0.55.

- adjust the brightness by adding a factor randomly chosen between -0.15 and 0.15 to each channel.

- zero padding that expands the size of the input image by four pixels horizontally and vertically, and then random cropping to the original size.

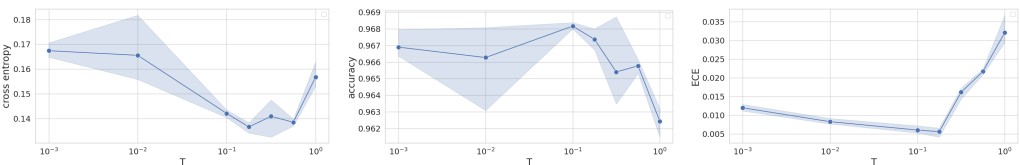

Figure 19: CPE on SVHN with data augmentation.

In Figure 19 and 20, we show the accuracy and ECE for the SVHN experiments with data augmentation of Section 4, Figure 3.

In Figure 21, we show the plots underlying Figure 5 for the data augmentation part.

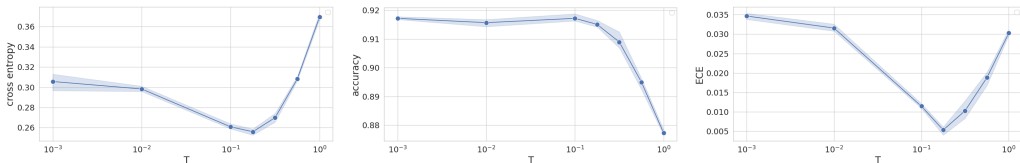

Figure 20: CPE on CIFAR-10 with data augmentation.

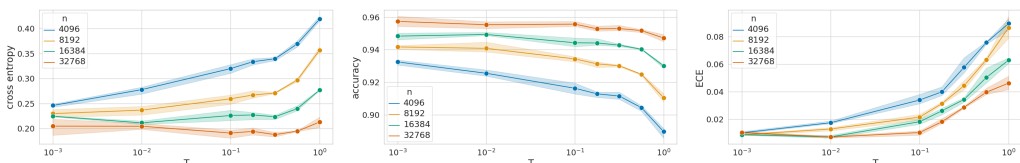

Figure 21: Data augmentation and sub-sampling. We additionally show the results for accuracy and ECE. Note a strong cold posterior effect

### E.5 Subsampling with decreasing number of gradient steps

We repeat the main subsampling experiment on SVHN of Section 4. This time we use a decreasing number of gradient steps. Results are in Fig. 22. In particular, the burn-in period is always 200 epochs. The cycle length is increased from 60 at $n = 8192$ to 100 at $n = 512$, and total number of epochs from 1100 to 1700. Therefore the number of gradient steps decreases, as both the cycle length and the total number of training epochs should be doubled every time that the dataset is halved.

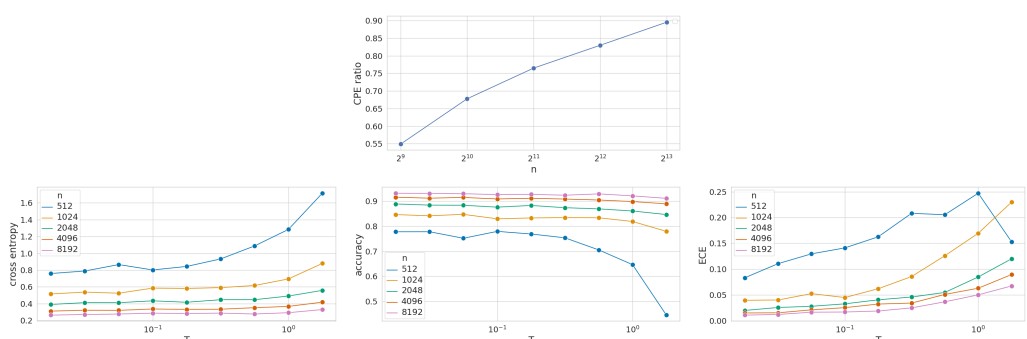

Figure 22: SVHN subsampling experiment with decreasing number of gradients steps.

### E.6 Experiments on CIFAR-10H

We repeat some of the experiment on CIFAR-10H by Aitchison (2020). Our results show that over-weighting the likelihood with respect to the prior is sufficient for the removal of the CPE. Furthermore, this weight does not equal the number of labellers, suggesting that the curation theory of Aitchison (2020) is not accurate enough to explain the CPE alone (as was already acknowledged by the author).

In Aitchison (2020), the authors state that the cold posterior effect might be due to the labelling process, based on consensus: each image is added to the dataset only if all $S$ labellers agree on one class. In particular, they say that if we had access to the $S$ original labels for each image, then we should not observe any cold posterior effect. They devise an experiment on CIFAR-10H (Peterson et al., 2019), in which all the (approximately 50) labels are given for each image. Our experiments show that over-weighting the likelihood with respect to the prior is sufficient to eliminate the CPE, and that the weight does not correspond to the number of labellers.

More formally, the dataset can be defined as $\mathcal{D} := \{(\boldsymbol{x}_i, \boldsymbol{y}_i)\}_{i=1}^n$, where $\boldsymbol{y}_i$ is a vector containing the *counts for each class*, i.e., its $j$-th element is the number of labellers that chose class $j$ Given a mini-batch of size $B$, the log-likelihood used in an SG-MCMC method, is the following:

$$\mathcal{L}_c := \frac{n}{B} \sum_{i=1}^B \boldsymbol{y}_i^T \log f_i. \tag{11}$$

This is very similar to the use of label smoothing in which the label vector is $\boldsymbol{y}_{ls} := \frac{1}{S}\boldsymbol{y}$. The corresponding mini-batch likelihood has the form:

$$\mathcal{L}_{ls} := \frac{n}{B} \sum_{i=1}^B (\boldsymbol{y}_{ls})_i^T \log f_i = \frac{n}{BS} \sum_{i=1}^B \boldsymbol{y}_i^T \log f_i. \tag{12}$$

Therefore the only difference between $\mathcal{L}_c$ and the "human-aware" label-smoothing loss $\mathcal{L}_{ls}$ is that the former is $S$ times stronger, and they are conceptually very similar. We will also consider "standard" label smoothing with parameter $\alpha \in (0, 1)$, :

$$\mathcal{L}_\alpha := \frac{n}{B} \sum_{i=1}^B \hat{\boldsymbol{y}}_i^T \log f_i, \tag{13}$$

where $\hat{\boldsymbol{y}}_i$ is $1 - \alpha$ in the position corresponding to the correct label and $\frac{\alpha}{C}$ otherwise.

For each of the likelihoods proposed above, we train a ResNet-20 on CIFAR-10H and evaluate on CIFAR-10 training set, as in Aitchison (2020). Regarding the SG-MCMC method, we use cyclical step size, adapting the code from Wenzel et al. (2020). We leave unchanged all the hyperparameters. The only exception is in the case we are using $\mathcal{L}_c$, where we reduce the learning rate by a factor $50$, the approximate number of labellers per image, as in Aitchison (2020). The reason for this reduction is that the likelihood gets $\approx 50$ times stronger due to the fact that each datapoint is labelled by $\approx 50$ labellers. We use 100 epochs as burn-in period and a cycle length of 50 epochs for a total of 1000 epochs. We use data augmentation. [10]

**Is it all about over-weighting the likelihood?**   We apply standard label smoothing (loss $\mathcal{L}_\alpha$) with $\alpha = 0.1$ to the one hot encoded labels. Then, we overweight the likelihood by a factor of $50$ and reduce the learning by the same factor (i.e. the approximate number of labellers). Results are in Figure 23.

Note that making the likelihood $S$ times stronger helps to eliminate the cold posterior effect. Note also that label smoothing plus likelihood over-weighting is equivalent to assuming that the labellers have assigned different labels.

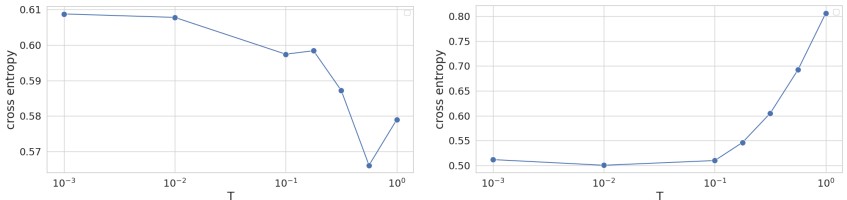

Figure 23: **left**: model with over-weighted likelihood with smoothed labels. **right**: model trained with standard label smoothing

**A smaller number of labellers is enough to alleviate the cold posterior effect**   The previous experiments have shown that over-weighting the likelihood alleviates the cold posterior effect. Here, we test whether weighting the likelihood by a smaller amount is sufficient to alleviate the cold posterior effect. In particular, we subsample a different number of $\tilde{S}$ labels for each image from the counts that are available in CIFAR-10H. We do this by considering the probabilities implied

---

[10]In Aitchison (2020) it is not explicitly stated that data augmentation have been used. However, we checked the code that they use (Zhang et al., 2020) and verified that data augmentation is indeed used there.

by the counts, and sampling $\tilde{S}$ times from this categorical distribution, for each image. Results are shown in Figure 24. Note that $T \geq 1$ is optimal for any value of $\tilde{S} \geq 10$ that we consider, and the cold posterior effect appears only for $\tilde{S} \leq 5$. This is strong evidence that there is not an exact correspondence between the number of labellers and the optimal temperature. What matters is to over-weight the likelihood, but even a significantly smaller weight can fix the cold posterior problem.

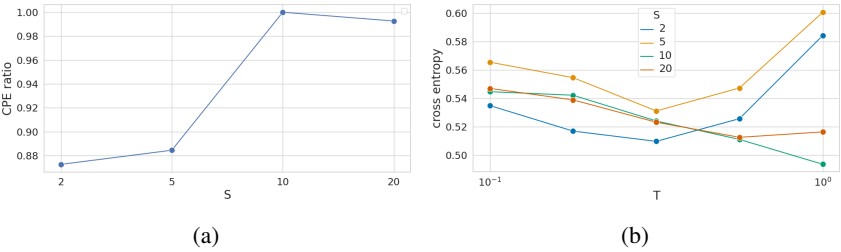

(a)                                            (b)

Figure 24: CPER and test CE for subsample of $\tilde{S}$ labellers from Cifar10H out of a total of $S \approx 50$ available labellers.