# OpenReview forum: "Disentangling the Roles of Curation, Data-Augmentation and the Prior in the Cold Posterior Effect"
_NeurIPS.cc/2021/Conference — NeurIPS 2021 Poster_

### Official Review · Reviewer_Qfnh · 2021-06-28

**Rating:** 7
**Confidence:** 4

**Summary:**

This paper studies the so-called cold-posterior effect (CPE), which refers to the poorly understood previous empirical observation that Bayesian posteriors sharpened by a temperature parameter T<1 perform better that the "correct" Bayesian posteriors corresponding to T=1. To me, the paper makes two nice contributions: first, a thorough discussion summarizing all previous explanations for the CPE from the literature; second, an insightful empirical analysis (or re-analysis) of the putative explanations and their impact on the CPE. The main finding was that the CPE does not have a single cause, but can be caused by any of: 1) dataset curation, 2) data augmentation, 3) bad priors. In particular, I think the paper implies that data augmentation and especially the prior have a larger role than previously appreciated, while the impact of dataset curation may not be that significant (although it's non-negligible in some setups, too).

**Limitations And Societal Impact:**

Yes.

**Main Review:**

PROS:
- A nice investigation of a relatively poorly understood phenomenon of the CPE, with novel insights that increase understanding, and I expect these results to be significant within the Bayesian DL community.
- Very clear presentation.

CONS:
- The paper identifies many causes for the CPE, but does not offer much practical guidance on how to avoid this issue in practice.
- The results are mainly empirical, without theory to back up the findings (I agree this might not be straightforward to come up with).


**Time Spent Reviewing:**

3

---

> ### Author Response · Authors · 2021-08-10
> **Response**
>
> We would like to thank the reviewer for their positive feedback.
>
> **1) “Practical guidance to avoid the CPE in practice”.**
>
> We agree that our paper does not directly resolve the CPE problem, but we do provide valuable insights on the amount of tempering that is needed based on what the practitioner knows about the problem at hand. In particular our results suggest that one should incorporate curation into the model if he or she knows that the dataset is heavily curated, but it is not needed for common benchmark datasets like SVHN and CIFAR-10, which have undergone a curation process but for which we do not experience any CPE (Figure 1). Secondly, our subsampling experiment has two important consequences: from the BNN community perspective, it highlights the importance of priors for BNNs, and this might drive future research in this direction. On the other hand, practitioners benefit from knowing that with smaller sample sizes, more tempering might be needed if one is unsure about the prior.
>
> **2) “Theory to back up empirical results”.**
>
> Little theory is known about the posterior over the network’s parameter in practical (finite) neural networks, and approximate methods are necessary to obtain samples from such posteriors. Our work focuses on understanding the need of tempering the posterior for such non-idealized settings, and therefore resorts to experimental verification. However, we do believe that our results provide valuable guidance in developing such a theory in the future.

---

> > ### Comment · Reviewer_Qfnh · 2021-08-19
> > **After author feedback**
> >
> > Thanks for your comments. I will keep my original score.

---

### Official Review · Reviewer_m6D5 · 2021-07-03

**Rating:** 7
**Confidence:** 4

**Summary:**

The paper presents a series of experiments on the cold-posterior effect, and argues that it is broadly prior-dominated,  rather than arising from e.g. the data curation hypothesis of Aitchison (2021).  While it is an interesting series of experiments that clearly deserve to be published, I have a number of concerns about the interpretation of the experiments.  It should be possible to deal with these during the rebuttal phase, at which point I'd be happy to increase my score.

**Main Review:**

General comments:
The figures need significant reformatting.  The fonts are tiny (basically unreadable from my printer).  And Figures should appear at the top, not in line.

Sec 2
=====

Worth noting that for large-scale ResNets (i.e. the model class we care about), Fortuin et al. 2021 show that more complex priors improve performance at all temperatures so the CPE remains basically unchanged (Fortuin et al. 2021; Fig. 7 bottom (CIFAR-10))

Sec 3
=====

Why isn't Figure 4 referenced here?  Why is Figure 4 placed after the Figures for other sections?  Except in exceptional circumstances, figures should just come in the order they are discussed.

Important to note an alterantive hypothesis consistent with the data.  In particular, if you believe the CPE is "variability driven" and works by suppressing stochasticity in the distribution over parameters.  In that case, there will be less CPE for more data because with more data the posterior is already highly concentrated.  If nothing else, if you have lots of data, then the posterior converges to the ML value of the parameters, and taking a cold posterior will have no effect at all, and this holds for any choice of prior!


Sec 4
=====

No CPE without data augmentation
--------------------------------

Very clean results without augmentation.

Conclusion that inaccurate inference is not necessary is correct.

"We can also conclude that the curation of SVHN and CIFAR-10 does not give rise to a CPE".  Only if the true model does not include augmentation.  But it is quite possible  that the true model incorporates data augmentation in the prior, as introduced in arxiv.org/abs/1808.05563, and as discussed further in contemporaneous work arxiv.org/abs/  2106.05586).  In particular, Bayes theorem only tells us that the true model should perform best.  But it doesn't say anything about the relative performance of             misspecified models.  So it is quite possible that a model with curation + augmentation (i.e. priors capturing data augmentation, as in arxiv.org/abs/1808.05563) is best,   while under the wrong model without augmentation, no curation beats curation.

CPE and synthetic curation
--------------------------

Correct.  In essence, including curation in the model makes the posterior more certain everywhere, which hurts performance on the ambiguous examples which would otherwise   be curated away.

Do standard Gaussian priors give too much weight to complicated hypotheses?
---------------------------------------------------------------------------

Isn't the more complicated distribution at T=1 better than the straight-lines at T=0.01?  The decision boundary should be more uncertain far from the data.

**Time Spent Reviewing:**

2

---

> ### Author Response · Authors · 2021-08-10
> **Response**
>
> We would like to thank the reviewer for their positive feedback and the valuable thoughts on our results. We will address the reviewer’s questions one by one.
>
> **1) “Figure formatting”.**
> Thank you for the pointer, we will reformat the figures as suggested.
>
> **-- Sec 2 --**
>
> **2) “more complex priors improve performance at all temperatures”.**
>
> We agree with the reviewer’s interpretation of Fortuin et al (2021). Different priors may eliminate the CPE for fully connected nets, but the CPE is still present when convolutional architectures are used (although different priors have substantial improvement across all temperatures). In Wilson et al. (https://arxiv.org/abs/2002.08791) the authors propose a “functional” view of the prior: as the individual weights have no clear interpretation, one should look at the macroscopic behaviour of the net, i.e. the functions a priori likely derived from propagating a signal through a neural network architecture with given prior. Under this view, it is reasonable that by maintaining the same Gaussian prior but changing the architecture, we observe different optimal temperatures.
>
> **-- Sec 3 --**
>
> **3) “Referencing Figure 4”.**
>
> We have used Section 3 to describe the theoretical motivation behind the subsampling experiment, leaving Section 4 entirely for experiments. Also, the CPE ratio - used to measure the strength of the CPE across all our figures -  is introduced only at the beginning of Section 4. The way the figures are currently placed, no jumps across Sections are required for the reader.
>
> **4) “Posterior concentration hypothesis”**
>
> We agree that, if enough data are provided, posterior concentration should happen regardless of the choice of the prior. This hypothesis is consistent with what we aim at testing here: for smaller dataset sizes, the prior choice will be more relevant, and tempering reduces the stochasticity in the parameters. A lower temperature for smaller datasets indicates that the prior might not have been chosen adequately, hence the need for tempering to “overcount” the data and reduce the importance of the prior.
>
> **-- Sec 4 --**
>
> **5) “No CPE without data augmentation”.**
>
> Very interesting thought! First of all, we would like to point out that in this discussion it is important to distinguish between “best performance” and “CPE”: a model could experience no CPE and perform worse than a model with a strong CPE but better performance at low temperatures. For instance, in our experiments using tempering + data augmentation (Figure 3) has better performance (at optimal temperature) than no data augmentation (Figure 1). Regardless of the relative performance gain of using curation in the model or not,  *what we test here is whether the CPE arises or not*. Figure 1 shows that on SVHN and CIFAR-10 (both of them have undergone a curation process) there is no need to incorporate curation in the model. This could indicate that those datasets are collected using a “milder” form of curation than what is described in Aitchison, 2020. If now data augmentation is used (Figure 3), the CPE arises. The fact that CPE arises also when data augmentation is incorporated in the model, as shown in the paper by Fortuin et al (2021) (Figure 2), could indeed indicate that one has to take curation into account, and our experiments cannot in general exclude it or confirm it. However, in light of our experiments on data augmentation, we would be surprised if the amount of tempering from curation becomes “visible” only after data augmentation is switched on. We also believe that there could be different ways to incorporate in the model those invariances that one wants to learn through data augmentation.
>
> We hope that this addresses the reviewer’s comments, but we are happy to continue the discussion if something remains unclear.
>
> We thank the reviewer for the pointers to the papers, and we are happy to add a discussion of these to our work.
>
>
> **6) “CPE and synthetic curation”.**
>
> We agree with this interpretation.
>
> **7) “Do standard Gaussian priors give too much weight to complicated hypotheses?”**
>
> Yes, we agree. It depends on what is meant by “better” here. In this case, the optimal separating curve is the line y=-x. Therefore a well-specified model + a good prior that confidently learns this line will have a better test cross entropy than the more complicated distribution at T=1. Here we hypothesize that a complex model like a neural network with Gaussian prior, will recover that optimal line with significantly more data than the tempered counterpart. We will adapt the text to make this clear.

---

### Official Review · Reviewer_CjSo · 2021-07-16

**Rating:** 7
**Confidence:** 3

**Summary:**

The paper aims at better understanding of the sources of the cold posterior effect (CPE) in Bayesian deep learning. It investigates the effects of some of the hypotheses proposed in the literature for this phenomenon, in particular: dataset curation, data-augmentation and bad priors. It examines each hypothesis via carefully designed experiments and empirically shows how CPE can arise from each of these sources in isolation, suggesting that there is no single simple cause.

**Limitations And Societal Impact:**

The only limitation I see is that the experiments of the paper are all classification tasks. I am wondering if any of the results would change for regression problems?

**Main Review:**

This paper does not provide a solution to the CPE problem. However, it aggregates some of the most prominent existing hypotheses and does a valuable effort to shed more light on the sources of the problem. The experiments are designed to disentangle different potential sources of the CPE problem.

Even though the paper does not propose a new hypothesis, its overview of the existing hypotheses and clarifications is valuable to the community.

The paper is very well-written and easy to follow and does a good job of covering the background and explaining its approach.

**Time Spent Reviewing:**

6

---

> ### Author Response · Authors · 2021-08-10
> **Response**
>
> We would like to thank the reviewer for their positive feedback.
>
> We agree that a significant part of our work consists in carefully testing existing hypotheses and whether / how they contribute to the CPE when considered in isolation.
> Our results suggest that the CPE may be a “symptom with many causes” (bad priors being one of them), hence we do not expect a simple “fix” for cold posteriors.
>
> **1) “Focusing on classification tasks”.**
>
> The reason why we have focused on classification tasks is that the CPE was identified in the classification setting in the majority of the recent literature, including the original paper (Wenzel et al. 2020, https://arxiv.org/pdf/2002.02405.pdf). Furthermore, the consensus theory proposed in Aitchison, 2020, https://arxiv.org/abs/2008.05912 that we test is not directly applicable in a regression setting, as there cannot be perfect agreement on labels over the reals.
>
> It is worth mentioning that the regression setting has been analyzed (in the context of infinite-width networks i.e. Gaussian processes) in Adlam et al., 2020, https://arxiv.org/abs/2008.00029. In a Bayesian linear regression setting with Gaussian prior, tempering the posterior over the parameters by T has the effect of scaling the covariance matrix by T. In particular, T < 1 causes a reduction in the predictive variance. In a classification problem (even logistic regression), the posterior becomes non analytically tractable and one has to resort to approximations (see e.g. chapter 3 of http://www.gaussianprocess.org/gpml/chapters/RW.pdf). However, the fact T < 1 implies “overcounting the data” (see for instance Wenzel et al (2020)) gives the intuition that tempering has a similar effect of uncertainty reduction for classification problems as well.
>
> We will add a short paragraph on the above mentioned work and other recent literature discussing the CPE in the regression setting.

---

> > ### Comment · Reviewer_CjSo · 2021-09-01
> > **Thank you for your reply**
> >
> > That you for taking the time to write the response. I will keep my score.

---

### Decision · Program_Chairs · 2021-09-27

**Decision:**

Accept (Poster)

**Comment:**

Another nice contribution to the ongoing community effort of understanding Bayesian deep learning.

All reviewers liked the paper.

Accept.